# Learning Search-Space Specific Heuristics Using Neural Network

**Liu Yu, Ryo Kuroiwa,[1] Alex Fukunaga,[2]**

[1]Department of Mechanical and Industrial Engineering, University of Toronto
[2]Graduate School of Arts and Sciences, The University of Tokyo
liuyu.ai@outlook.com, mhgeoe@gmail.com, fukunaga@idea.c.u-tokyo.ac.jp

## Abstract

We propose and evaluate a system which learns a neural-network heuristic function for forward search-based, satisficing classical planning. Our system learns distance-to-goal estimators from scratch, given a single PDDL training instance. Training data is generated by backward regression search or by backward search from given or guessed goal states. In domains such as the 24-puzzle where all instances share the same search space, such heuristics can also be reused across all instances in the domain. We show that this relatively simple system can perform surprisingly well, sometimes competitive with well-known domain-independent heuristics.

## 1 Introduction

State-space search using heuristic search algorithms such as GBFS is a state-of-the-art technique for satisficing, domain-independent planning. Search performance is largely determined by the heuristic evaluation function used to decide which state to expand next. Heuristic function effectiveness for domain-independent planning depends on the domain, as different heuristics represent different approaches to exploiting available information. Designing heuristics which work well across many domains is nontrivial, so learning-based approaches are an active area of research.

In one setting for learning search control knowledge for planning (exemplified by the Learning Track of the IPC), a set of training problem instances (and/or a problem instance generator) is given, and the task is to learn a *domain-specific* heuristic for that domain. Previous work on learning heuristics and other search control policies (e.g., selection among several search strategies) in this setting include (Yoon, Fern, and Givan 2008; Xu, Fern, and Yoon 2009; de la Rosa et al. 2011; Garrett, Kaelbling, and Lozano-Pérez 2016; Sievers et al. 2019; Gomoluch, Alrajeh, and Russo 2019). Another type of setting seeks to learn domain-independent planning heuristics, which generalize not only to domains used during training, but also to unseen domains (Shen, Trevizan, and Thiébaux 2020; Gomoluch et al. 2017).

Inter-instance speedup learning, or "on-line learning", is a setting where only one problem instance (no training instances or problem generator) is given, and the task is to solve that instance as quickly as possible. Speedup learning within a single problem solving episode is worthwhile

if the *total time spent by the solver (including learning)* is faster than the time required to solve the problem using other methods. Previous work on on-line learning for search-based planning includes learning decision rules for combining heuristics (Domshlak, Karpas, and Markovitch 2010) and macro operator learning (Coles and Smith 2007).

On-line learning can be used to learn an *instance-specific heuristic*. Previous work on instance-specific learning includes bootstrap heuristic learning (Arfaee, Zilles, and Holte 2011), as well as LHFCP, a single-instance neural network heuristic learning system (Geissman 2015). Instance-specific learning can be generalized to *single search space learning*, where many problem instances share a single search space. For example, all instances of the 15-puzzle domain share the same search space – different instances have different initial states, all on the same connected state space graph. Thus, a learned heuristic function which performs well for one instance of the 15-puzzle can be directly applied to other instances of the domain.

We propose and evaluate SING, a neural network-based instance-specific and single search space heuristic learning system for domain-independent, classical planning. SING is closely related to LHFCP, an approach to supervised learning of heuristics which generates training data using backward search (2015). Given a PDDL problem instance $I$, LHFCP learns a heuristic $h_{nn}$ for $I$. To generate training data for $h_{nn}$, LHFCP performs a series of backward searches from a goal state of $I$ to collect a set of states and their approximate distances from the goal. After training, $h_{nn}$ is used as the heuristic function by GBFS to solve $I$. This does not require any additional training instances as input, nor pre-existing heuristics to bootstrap its performance. However, LHFCP performed comparably to blind search (Geissman 2015), so achieving competitive performance with this approach remained an open problem.

SING expands upon this basic approach in several ways: (1) improved backward search space using either (a) explicit search with inferred inverse operators or (b) regression, (2) depth-first search (vs. random walk), (3) boolean state representation, and (4) relative error loss function. We experimentally evaluate SING for learning domain-specific heuristics for domains where instances share a single state space, and show performance competitive with the Fast Forward heuristic ($h_{ff}$) (Hoffmann and Nebel 2001), and the land-

mark count heuristic ($h_{\text{lm}}$) (Hoffmann, Porteous, and Sebastia 2004) on several domains. We also evaluate SING as an instance-specific heuristic learner, and show that the learned heuristics consistently outperforms blind search on a broad range of standard IPC benchmark domains, and performs competitively on some domains, even when the learning times are accounted for within the time limit.

## 2 Preliminaries and Background

We consider domain-independent classical planning problems which can be defined as follows. A SAS+ planning task (Bäckström and Nebel 1995) is a 4-tuple, $\Pi = \langle V, O, I, G, \rangle$, where $V = x_1, ..., x_n$ is a set of state variables, each with an associated finite domain $Dom(x_i)$; A state $s$ is a complete assignment of values to variables. $\mathcal{S}$ is the set of all states. $O$ is a set of actions, where each action $a \in O$ is a tuple $(pre(a), eff(a))$, where $pre(a)$ and $eff(a)$ are sets of *partial* state variable assignments $x_i = v, v \in Dom(x_i)$; $I \in S$ is the initial state, and $G$ is a partial assignment of state variables defining a goal condition ($s \in S$ is a goal state if $G \subset s$). A plan for $\Pi$ is a sequence of applicable actions which when applied to $I$ results in a state which satisfies all goal conditions. Search-based planners seek a path from the start state to a goal state using a search algorithm such as best-first search guided by a heuristic state evaluation function.

A natural approach to learn heuristic functions for search-based planning is a supervised learning framework consisting of the following stages: (1) **Training Sample Generation**: generate many state/distance pairs which will be used as training data. (2) **Training**: Train a heuristic function $h$ which predicts distances from a given state to a goal. (3) **Search**: Use $h$ as the heuristic evaluation function in a standard heuristic search algorithm such as GBFS. This paper focuses mostly on stage (1), training data generation.

Ferber et al. (2020) investigated an approach where the training data was generated using forward search from the start states of training instances. They perform random walks (200 steps) from the initial state, and from each step visited in the random walk, they perform a forward search (a "teacher search" using a heuristic such as $h_{\text{ff}}$) to a goal in order to find the distance to the goal. If the teacher search finds a path to the goal, the states on the path as well as the distance-to-goal for the states on the path are added to the training data. This approach can be practical for shared search space heuristic learning, where the costs of the teacher searches can be amortized among many instances on the same search space. However, this is not practical for satisficing, single instance heuristic learning where there is only one problem solving episode, as requiring forward search to the goal in order to gather training data obviates the need to learn a heuristic for that particular instance.

An alternative approach to generating training data uses backward search from the goal. A backward search starting at a goal state (provided in or guessed/derived from the problem specification) is performed, storing encountered states and their (estimated) distances from thoe goal as the training data. Arfaee, Zilles, and Holte (2011) used this approach in a bootstrap system for heuristic learning, which starts with

a weak neural net heuristic $h_0$ and generates increasingly more powerful heuristics by using the current heuristic to solve problem instances, using states generated during the search as training data for the next heuristic improvement step. If $h_0$ is too weak to solve training set problems, they generate training data by random walks from the goal state to generate easy problem instances that can be solved by $h_0$. Lelis et al. (2016) proposed BiSS-h, an improvement which uses a solution cost estimator instead of search for training data generation. Arfaee et al. and Lelis et al. evaluated their work on domain-specific solvers for the 24-puzzle, pancake puzzle, and the Rubik's cube.

Geissmann (2015) investigated a backward-search approach to training data generation for domain-independent classical planning. His system, LHFCP, uses backward search to generate training data for learning a neural network heuristic function which estimates the distance from a state to a goal. To generate training data, LHFCP performs backward search (random walk) in an explicit search space. It generates the start state for backward search by generating a state which satisfies the goal conditions, with values unspecified by the goal condition filled in randomly. LHFCP relies on the operators in the original (forward) problem to perform backward search. Search using the heuristics learned by LHFCP across a wide range of IPC domains performed comparably to blind search (Geissman 2015). Geissmann also investigated a variation of LHFCP which applied BiSS-h to classical planning but reported poor results, attributed to difficulties in efficiently implementing BiSS for classical planning. Thus, a successful backward-search based approach to training data generation for domain-independent classical planning remained an open problem.

## 3 SING: An Improved, Backward-Search Based Heuristic Learning System

We describe **SI**ngle search space **N**eural heuristic **G**enerator (SING), a system which learns single-search space heuristics for domain-independent planning. SING learns a heuristic function $h_{nn}(s)$, which takes as input a vector representation of a state $s$, and returns a heuristic estimate of the distance from $s$ to a closest goal. SING is implemented on top of the Fast Downward planner (FD) (Helmert 2006).

SING uses backward search to generate training data, similar to LHFCP, but incorporates several significant differences in the state representation, backward search space formulation, and backward search strategy. Below, we describe each of these in details:

### 3.1 State Representation

The input to $h_{\text{nn}}$ is a vector representation of a state. LHFCP used a multivalued SAS+ vector representation of the state, which is a natural representation to use, as FD uses the SAS+ representation internally.

Another natural representation for the vector input to $h_{\text{nn}}$ is based on the STRIPS propositional representation of the problem. A STRIPS planning task (Fikes and Nilsson 1971) is a 4-tuple, $\Pi = \langle F, I, G, A, \rangle$ where $F$ is a set of propositional facts, $I \in 2^F$ is the initial state, $G \in 2^F$ is a set of

goal facts, and $A$ is a set of actions. Each action $a \in A$ has preconditions $pre(a)$, add effects $add(a)$, and delete effects $del(a)$, which are sets of facts. A state $s \in 2^F$ is set of facts, and $s$ is a goal state if $G \subseteq s$. Given a state $s \in 2^F$, $a$ is applicable iff $pre(s) \subseteq s$. After applying $a$ in $s$, $s$ transitions to $s \cup add(a) \setminus del(a)$. A plan for $\Pi$ is a sequence of applicable actions which make $I$ transition to a goal state.

The STRIPS representation corresponds directly to the classical planning subset of the standard PDDL domain description language, as PDDL uses boolean facts to represent the world state. In the SAS+ representation used by FD, each possible value of a variable represents a mutually exclusive set of facts in the underlying propositional problem. Each variable-value pair in FD represents a fact, negation of a fact, or negation of all facts represented by other values in the variable. Preconditions and effects of actions are also represented as the set of variable-value pairs. Since the variable/value naming conventions used in the SAS+ generated by the FD PDDL-to-SAS+ translator, conversion between the SAS+ finite-domain representation and the STRIPS propositional state representation is easy. Thus, $h_{\mathrm{nn}}$ can use either the boolean (STRIPS) or multivalued (SAS+) state vector representation as input during training and search.

Since each input bit corresponds to a fact in the boolean encoding, it may enable a more accurate $h_{\mathrm{nn}}$ state evaluation function to be learned than the SAS+ multi-valued encoding. On the other hand, SAS+ encodings are more compact, which can significantly reduce the dimensionality of the state representation, which can result in faster NN evaluation, speeding up the search process. Thus, the choice of state vector representation poses a tradeoff between $h_{\mathrm{nn}}$ evaluation accuracy and $h_{\mathrm{nn}}$ evaluation speed, and SING can use either the multivalued SAS+ vector representation or the STRIPS boolean vector representation.

## 3.2 Search Space and Operators for Training Sample Generation

The task of training sample generation is to collect a training set $T = \{(s_1, e_1), ..., (s_r, e_r)\}$ a set of $r$ states and their (estimated) distance to a goal. The basic idea is to repeatedly start at a goal $g$ and generate a sequence of states heading away from it (using a directed search or random walk), adding such states to the training data.

In some search problems such as the sliding tiles puzzle, backward search is relatively straightforward as the goal state is given explicitly as input to the problem, and the operators available for the forward problem are sufficient to solve the backward problem.

In domain-independent planning, backward search based training sample generation poses several issues. First, a goal condition, possibly satisfied by many goal states, is given instead of an explicit, unique goal state, so in general, it is not possible to simply "search backward from the goal state". Second, in general, the operators for the forward problem are not sufficient for backward search. LHFCP generates a start state for backward search by generating a state which satisfies the goal conditions, with values unspecified by the

goal condition filled in randomly. It relies on the operators in the original (forward) problem to perform backward search.

SING incorporates two approaches to backward search for training sample generation: (1) backward explicit search using derived inverse operators, and (2) regression.

**Explicit Backward Search with Derived Inverse Operators**  As in LHFCP, a candidate start state for backward search is generated by first generating a partial state which satisfies all conditions in the goal condition, and then randomly assigning values to variables whose values are unspecified in the goal condition. Such a randomly generated candidate start state $s$ might be invalid and unexpandable, i.e., no backward operators (see below) can be applied to $s$. In that case, we simply generate another candidate state. This random initialization is performed for each backward sampling search.

For the search operators, one simple approach is to use the same set of actions as for forward search, as in LHFCP (Geissman 2015). However, this fails in domains where actions are not invertible such as `visitall`.

Thus, operators for the backward search must be derived from the forward operators. Since preconditions and effects are represented as a set of variable-value pairs in FD, one naive method to generate inverse actions is to swap values of variables which appear in both preconditions and effects. Other variable-value pairs in preconditions and effects are treated as preconditions in the inverse action, because they must hold after application of the action. However, the inverse action does not change values of variables which appear in the original effects but not in the original preconditions. To address this issue, we use information available in the STRIPS formulation of the problem (as explained in Section 3.1, conversion among the PDDL problem description, its STRIPS formulation and the SAS+ formulation used internally by Fast Downward is straightforward). For action $a$, we generate an inverse action $a'$ such that $pre(a') = (pre(a) \cup add(a)) \setminus del(a)$, $add(a') = del(a)$, and $del(a') = add(a)$. We identify variable-value pairs which represent propositions as add effects, and pairs which represent negation of facts as delete effects.

**Regression**  Another approach to backward search is regression. In backward search using regression, we use the modified SAS+ representation by Alcázar et al. (2013). An action $a$ is applicable to a state $s$ if $add(a) \cap s \neq \emptyset \wedge del(a) \cap s = \emptyset$. If an action $a$ is applied to a state $s$, $s$ transitions to a state $s' = (s \setminus add(a)) \cup pre(a)$. Normally, SAS+ variables represent mutex groups of the corresponding STRIPS propositions. In regression planning with SAS+, each variable has an additional, *undefined* value. The starting node in regression space is the goal state, where variables unspecified by the goal condition have *undefined* values. When an action $a$ is applied to a state $s$, if a variable $v$ is included in $add(a)$ but not in $pre(a)$, $v$ is set to *undefined*.

When generating training data, a bit vector representation of states needs to be generated (Section 3.1). When converting the SAS+-based representation used by Fast Downward into a bit vector, unlike the other possible state values in the mutex group, undefined values are not explicitly represented

in the bit vector. For example, suppose a state variable $x$ in regression search has 2 possible actual values, $v_1$ and $v_2$, as well as "undefined". In the bit vector representation output for use as training data, $x$ is represented by 2-bits, where the first bit represents $x_1$, and the second bit represents $x_2$, and there is no explicit third bit for the undefined value.

**Regression vs. explicit search spaces** The choice of regression vs. explicit spaces depends on the domain. Although regression is in a sense the "correct" way to perform backward search, the backward branching factor in regression space is very large in many domains. On the other hand, while explicit backward spaces sometimes have much smaller branching factor than regression, goal generation has risks. First, goal generation might fail to find a goal. Also, generated goals might not be true goal states reachable from the start state, and states reachable from such incorrect states are also unreachable from the start state. Such cliques (unreachable from the start state) can cause backward search to yield few or no states for training data.

However, although the training data may include states which are unreachable from the start state, these may nevertheless be useful for learning an effective $h_{nn}$ which evaluates "real" states during search, somewhat similar to how synthetic data generated by the adversary during training is useful for learning networks which correctly classify real data in GAN learning (Goodfellow et al. 2014).

### 3.3 Backward Search Strategy

Given a start state for backward search (corresponding to a goal in the forward search space), we seek a set of training states $T$ which are relatively far from goal $g$ but with a reasonable estimate of their distance from $g$ for training $h_{nn}$. Breadth-first search (BFS) from $g$ could be used to generate states $T$ for which $c(s, g)$, the exact distances from $s \in S$ to $g$ (assuming unit-cost domains) are known, but would limit the training data to states which are very close to $g$. We need a search algorithm which can go much further from $g$ than BFS, and for which the number of steps in the (inverted) path from $s \in T$ to $g$ is an approximation of $c(s, g)$.

One natural sampling/search strategy is random walk, as in LHFCP (Geissman 2015). The number of steps from $g$ at which $s$ is encountered is used as an estimate of the true distance from $g$ to $s$. Although random walk is fast, distance estimates from random walk may be inaccurate if cycles are not detected. Loop detection can be implemented easily using a hash table, but in domains with many cycles, it can be difficult to sample nodes far from $g$ if the random walk is restarted whenever a previously visited node is generated.

Therefore, we use depth-first search (DFS) to extend a path from $g$, using the depth at which $s$ is encountered is used as an estimate of the true distance from $g$ to $s$, and all generated states are added to $T$. Random tie-breaking among $s$ is used to select the nodes among successors of $s$, $Succ(s)$, to expand. A hash table is used to prune duplicate nodes and prevent cycles. In domains with many cycles and dead ends, by backtracking (instead of restarting search) when a duplicate is detected, DFS can potentially sample more states which are further from $g$ than random

walk. The best choice of sampling search strategy depends on the domain. In some domains, DFS generates more accurate samples than random walk due to duplicate detection and backtracking, while in other domains DFS may incur large overheads due to backtracking and Random walk allows faster searches.

In the experiments below, during training data generation we perform *nsearches* backward searches, stopping each search after *nsamples* states are collected, i.e., *nsearches* $\times$ *nsamples* states are collected.

### 3.4 Neural Network Architecture

We use a standard feedforward network for $h_{nn}$, using the ReLU activation function. Each layer is fully connected to the next layer. The input layer takes the state vector representing a state $s$ as input. As discussed in Section 3.1, the state vector is either a boolean vector for the STRIPS representation of the problem instance, or a multivalued vector for the SAS+ representation of the instance, so the number of inputs is the same as the length of the state vector ($|F|$ for STRIPS propositional representation, $|V|$ for SAS+ multivalued representation). The output layer is a single node which returns $h_{nn}(s)$, the heuristic evaluation value of state $s$. Since $h_{nn}$ will be called many times as the heuristic evaluation function for best-first search, a small network enabling fast evaluation is desirable.

PyTorch 1.2.0 is used for training $h_{nn}$, but for search, we use the Microsoft ONNX Runtime 0.4.0 to evaluate $h_{nn}$. Both training and search use a single CPU core. Due to the simple network architecture as well as accelerated evaluation using the ONNX Runtime, $h_{nn}$ can be evaluated relatively quickly, significantly faster than $h_{ff}$ on most IPC domains, (see node expansion rates in Table 2).

### 3.5 Loss Function

Previous work on learning neural nets for classical planning used the standard Mean Square Error (MSE) regression loss function (Geissman 2015; Ferber, Helmert, and Hoffmann 2020). Instead of MSE, we use a prediction *relative error* sum loss function, $f_{loss} = \sum_i abs(\hat{y}_i - y_i)/(y_i + 1)$, which is the sum of the *relative* error of the predicted ($\hat{y}_i$) values compared to the training data ($y_i$).

## 4 Evaluation: Domain-Specific Heuristic Learning on Shared Search Spaces

In domains where multiple instances share the same space, it is possible to learn reusable $h_{nn}$ networks that can be used across many instances, so the cost of learning a heuristic can be amortized across instances. For example, all instances of the $N$-puzzle (for a particular value of $N$) share the same search space.

We evaluated SING as a shared search space, single model learner on the following PDDL domains:

- `24-puzzle`: PDDL encodings of the standard 50-instance benchmark set from (Korf and Felner 2002)

- `35-puzzle`: 50 randomly generated instances

| name | state vector | backward space | rev. search | inversion | NN # of hidden | NN nodes hidden | samples # |
|------|-------|---------|--------|-----------|--------|--------|----------|
| C2 | boolean | regression | DFS | yes | 1 | 16 | $10^5$ |
| C3 | SAS+ | explicit | rand. walk | yes | 1 | 16 | $10^5$ |
| C4 | boolean | explicit | DFS | yes | 4 | 64 | $10^5$ |
| C5 | boolean | explicit | DFS | yes | 1 | 16 | $4 \times 10^5$ |
| SING/L | SAS+ | explicit | rand. walk | no | 1 | 16 | $10^5$ |

Table 1: SING configurations used in experiments. "state vector": vector representation of states. "backward space": search space for training data generation backward search. "rev. search" : search strategy for Training data generation backward search. "NN # of hidden": # of hidden nodes in $h_{nn}$. "NN nodes hidden": # of nodes per hidden layer. "samples #" : # of sample states collected in the training data collection phase using the sampling search. C2, C3 and C4 perform 500 searches, with a limit of 200 samples/search ($10^5$ samples). C5 performs 800 searches, with a limit of 500 samples/search ($4 \times 10^5$ samples).

- `blocks25`: 100 `blocks` instances with 25 blocks generated using the generator from (Hoffmann 2002)

- `pancake`: 100 randomly generated instances with 14 pancakes.

For each domain above, we ran the learning phase (training data generation and $h_{nn}$ training) *once* to learn a heuristic $h_{nn}$ for the domain. For `24-puzzle`, we used the C4 configuration (Table 1 shows configuration details), and training data generation took 7 seconds and training took 61 seconds. For `blocks25`, we used the C5 configuration, training data generation took 502 seconds and training took 228 seconds. For `pancake`, we used the C4 configuration, training data generation took 21 seconds and training took 377 seconds.

Note that for these 3 domains, we tried several SING configurations (i.e., manual tuning) and report the results for the best configuration. We are currently investigating automated tuning (hyperparameter optimization) to optimize the best configuration for a given domain.

Table 2 and Figures 1-2 compare the coverage, node expansions, and runtime (on solved instances) of GBFS using $h_{nn}$, $h_{ff}$, $h_{lm}$ with a 30 min time limit per instance and 8GB RAM limit using an Intel(R) Xeon(R) CPU E5-2680 v2.

$h_{nn}$ had or tied for the highest coverage on all 4 domains. On `blocks25` and `pancake`, $h_{nn}$ had the highest coverage. On `24-puzzle`, `35-puzzle` and `pancake`, $h_{nn}$ had the lowest median run time. Thus, $h_{nn}$ achieved competitive performance on all of these domains compared to both $h_{ff}$ and $h_{lm}$ in this shared search space evaluation setting. Note that while $h_{nn}$ and $h_{ff}$ expanded a comparable number of nodes, $h_{nn}$ had a significantly higher median node expansion rate than $h_{ff}$ resulting in faster runtimes.

Figure 3 compares heuristic accuracy ($h$-value minus true distance) for a set of 4400 states for $h_{nn}$, $h_{ff}$, $h_{lm}$, and $h_{gc}$ (goal count). For states with true distance $\leq 30$ from the goal state, $h_{nn}$ is fairly accurate. This accuracy and the fast evaluation speed due to the simple neural network enables efficient, fast search.

## 5 Evaluation: Instance-Specific Learning

In Section 4, we evaluated SING for learning domain-specific heuristics which could be reused on many instances sharing the same search space, so the evaluation focused on search time, assuming that the time spent learning $h_{nn}$ can be amortized across multiple instances.

Next, we evaluate SING as an instance-specific learner in an IPC Satisficing track setting, where SING is given 30 minutes total for all phases, including learning (including training data collection and training) and search. Each run of SING starts from scratch – *nothing is reused across instances, learning costs are not amortized, and the heuristic is learned specifically for solving a given instance once.*

We evaluate SING on a large set of standard benchmarks from the IPC, all with unit-cost actions. All runs were given a total 30 minutes for time limit both learning and search (i.e., includes training data collection, training, and search using $h_{nn}$) and 8GB RAM per instance. We evaluated the SING configurations in Table 1. As baselines for comparison, we also evaluated blind search, the goal count heuristic ($h_{gc}$), the Fast Forward heuristic ($h_{ff}$) (Hoffmann and Nebel 2001), and the landmark count heuristic ($h_{lm}$) (Hoffmann, Porteous, and Sebastia 2004). As an additional baseline we also evaluate SING/L, a configuration of SING which is very similar to LHFCP (Geissman 2015) (see Table 1. This configuration is the same as C3, except that instead of the derived inverse operators (Section 3.2), SING/L uses only the actions available in the forward model.

Table 3 shows the coverage results (# of instances solved). SING configurations C2, C3, C4, C5 significantly outperform blind search, showing that SING successfully learned some useful heuristic information.

The SING/L (LHFCP) configuration performed comparably to blind search, consistent with the results in (Geissman 2015). Configuration C3, which differs from SING/L only in that action inversion is used, has much higher coverage than SING/L, showing the effectiveness of action inversion.

C2 outperforms $h_{ff}$ on 5 domains and outperforms $h_{lm}$ on 2 domains. C3 outperforms $h_{ff}$ on 5 domains and $h_{lm}$ on 1 domains. C4 outperforms $h_{ff}$ on 4 domains, and C5 outperforms $h_{ff}$ on 3 domains. Thus although none of the SING configurations are competitive with $h_{ff}$ and $h_{lm}$ with respect to overall coverage, these results indicate that there are some domains where competitive performance can be obtained with a 30 minute limit, including the time to learn an instance-specific heuristic function entirely from scratch without a teacher.

## 6 Ablation Study

To understand the relative impact of each of the new components of SING compared to LHFCP, we performed an ablation study comparing the following configurations:

(1) C5': Configuration C5 (Table 1) with fewer training samples (100k instead of 400k), (2) C5'/rw : same as C5', except using random walk instead of DFS, (3) C5'/sas : same as C5', except using SAS+ instead of boolean state representation, (4) C5'/reg : same as C5', except using regression instead of explicit search state, (5) C5'/orig : same as C5',

|  | coverage rate | | | median #expansions | | | median #exp. per second | | | median runtime | | |
|---|---|---|---|---|---|---|---|---|---|---|---|---|
|  | $h_{nn}$ | $h_{ff}$ | $h_{lm}$ | $h_{nn}$ | $h_{ff}$ | $h_{lm}$ | $h_{nn}$ | $h_{ff}$ | $h_{lm}$ | $h_{nn}$ | $h_{ff}$ | $h_{lm}$ |
| 24-puzzle | **100.0** | **100.0** | **100.0** | **5,514** | 9,232 | 67,859 | 10,649 | 3,862 | **39,633** | **0.52** | 2.31 | 1.59 |
| 35-puzzle | **100.0** | **100.0** | **100.0** | 122,463 | **57,045** | 1,650,552 | 9,313 | 3,749 | **74,487** | **12.95** | 15.20 | 21.86 |
| blocks | **84.0** | 73.0 | 83.0 | 353,856 | 332,974 | **33,658** | 14,926 | 2,830 | **24,054** | 26.07 | 126.14 | **1.56** |
| pancake | **100.0** | 48.0 | **100.0** | **74,873** | 324,925 | 1,620,030 | 25,261 | 912 | **134,248** | **2.93** | 347.55 | 10.60 |
| Average | **96.0** | 80.2 | 95.8 | **139,177** | 181,044 | 843,025 | 15,038 | 2,838 | **68,106** | 10.61 | 122.80 | **8.90** |

Table 2: Domain-specific heuristics: Reusing a single learned model across many instances of the same shared search space domain. The (sampling, training) times were (28s, 210s) for `24-puzzle`, (276s, 764s) for `35-puzzle`, (502s, 228s) for `blocks25`, and (21s, 377s) for `pancake`.

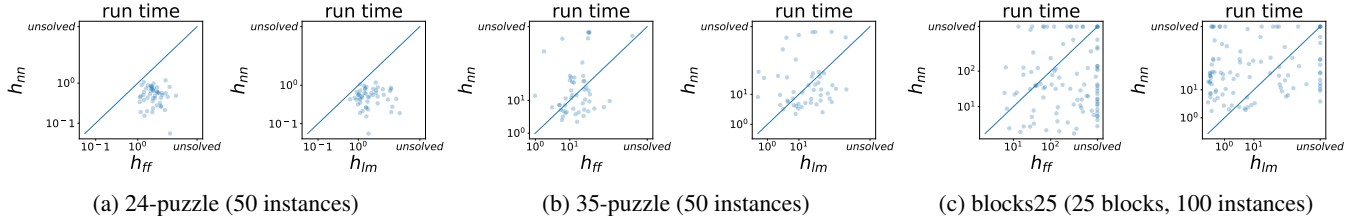

(a) 24-puzzle (50 instances)      (b) 35-puzzle (50 instances)      (c) blocks25 (25 blocks, 100 instances)

Figure 1: Runtime (seconds) for `24-puzzle`, `35-puzzle`, and `blocks25`. $h_{nn}$ vs. $h_{ff}$ and $h_{lm}$.

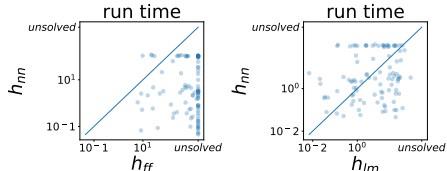

Figure 2: Runtime (seconds) for `pancake` (14 pancakes, 100 instances). $h_{nn}$ vs. $h_{ff}$ and $h_{lm}$.

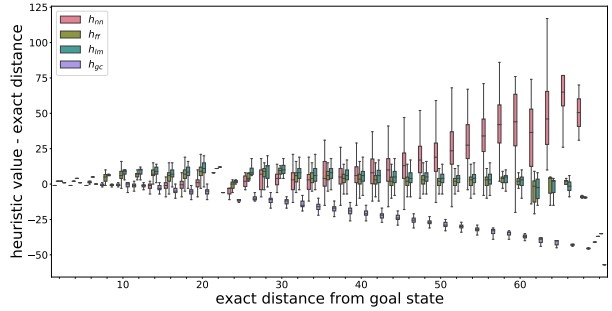

Figure 3: `24-puzzle` Heuristic accuracy: $h_{nn}$, $h_{gc}$, $h_{ff}$, $h_{lm}$

except using original operators only (no action inversion), and (6) C5'/mse : same as C5', except using MSE instead of the relative error sum loss function for NN training.

All configurations were run with a 30min, 2GB limit on the same IPC instances used in the above experiment. The coverages of the configurations were 604 for C5', 563 for C5'/rw, 481 for C5'/sas, 651 for C5'/reg, 420 for C5'/orig, and 559 for C5'/mse. This shows that the use of DFS in backward search, the use of boolean state representation, the use of action inversion, and the use of relative error sum loss function all have a significant positive impact on per-

formance.

On the other hand, the effect of using regression vs explicit state search for the backward search during training data generation is highly domain-dependent, with regression performing better on some domains and explicit search on others, as can be seen by comparing configurations C2 (which is the same as C5'/reg) vs. C5 in Table 3.

# 7 Related Work

A broad survey of learning for domain-independent planning is (Celorrio et al. 2012). Satzger and Kramer (2013) developed a neural network based, domain-specific heuristic for classical planning. They used random problem generators to create instances for training the neural network. Their training process also relies on the use of an oracle (the FD planner with an admissible heuristic) to provide true distance from a state to a goal.

Shen et al. (2020) proposed an approach to learning domain-independent (as well as domain-dependent) heuristics using Hypergraph Networks. They showed that it was possible to successfully learn domain-independent heuristics which performed well even on domains which were not in the training data. As this approach uses a hypergraph based on the delete relaxation of the original planning instance, it is quite different from the minimalist approach taken in SING, which does not use any such derived features and uses only the raw state vector. The training data generation method is forward search based, similar to the forward approach of Ferber et al. described in Section 2 (Ferber, Helmert, and Hoffmann 2020). In addition, while their work focuses on generalization capability and search efficiency (node expansions) across domains, with runtime competitiveness left as future work, our work seeks to achieve runtime competitiveness using a simple NN architecture.

Random-walk sampling of the search space of determin-

istic planning problems for the purpose of learning a control policy for a reactive agent was proposed in (Fern, Yoon, and Givan 2004). This differs from SING in that SING learns a heuristic function which estimates distances to a goal state and and guide search (GBFS), instead of a reactive policy.

There is also a rapidly growing body of work on learning neural network based policies for probabilistic domains (c.f., (Toyer et al. 2018; Issakkimuthu, Fern, and Tadepalli 2018; Groshev et al. 2018; Bajpai, Garg, and Mausam 2018; Garg, Bajpai, and Mausam 2019)), which is also related to learning heuristic evaluation functions for deterministic domains.

## 8 Conclusion

We investigated a supervised learning approach to learning a heuristic evaluation for search-based, domain-independent classical planning, where the training data is generated using backward search. Although LHFCP, a previous system, followed the same basic approach, it was performed comparably to blind search. SING pushes this approach much further using (1) backward search for training data generation using regression, as well as derived inverse operator for explicit space search, (2) DFS-based backward search for training data generation, (3) a propositional input vector representation, and (4) a relative error loss function.

We showed that SING can achieve performance competitive with $h_{ff}$ and $h_{lm}$ on several domains, both in shared search space scenarios where heuristics can be reused across domains, as well as single-instance learning where both learning and search using the learned heuristic must be performed within a given time limit.

SING is a relatively simple, minimalist system. SING uses *only* a PDDL description of a single problem instance as input. No additional problem generators or training instances are used. Learning is from scratch, and unlike the forward search based training data generation approach investigated by (Ferber, Helmert, and Hoffmann 2020), SING does not use any standard heuristics during training data generation. It uses a very simple feedforward neural network architecture, with no feature engineering. The only "features" used by SING are the raw state vectors. SING does not exploit any structures used by standard classical planning heuristics such as delete relaxations and causal graphs in either the learning or the search phases. Previous work used features derived/extracted from human-developed heuristics such as $h_{ff}$ and explored how learning could be used to exploit such features in new ways (Yoon, Fern, and Givan 2008; Xu, Fern, and Yoon 2009; de la Rosa et al. 2011; Garrett, Kaelbling, and Lozano-Pérez 2016; Shen, Trevizan, and Thiébaux 2020). By pushing the performance envelope for a more minimal approach our results provide a baseline for future work on heuristic learning using more sophisticated features and methods.

As discussed in Section 3.2, explicit backward search (as opposed to regression) for training data generation can generate states which are not reachable from the start state. Nevertheless, our results show that SING configurations which use explicit backward search perform quite well on some domains. In future work, we will investigate in detail how unreachable states in the training data affect the quality of the learned heuristic and the performance of the (forward) search using the learned heuristic.

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

| | blind | gc | hff | hlm | SING/L (lhfcp) | c2 | c3 | c4 | c5 |
|---|---|---|---|---|---|---|---|---|---|
| agricola | 0 | 0 | 10 | **16** | 1 | 13 | 12 | 11 | 9 |
| airport | 23 | 35 | **36** | 33 | 1 | 21 | 13 | 14 | 17 |
| barman | 0 | 0 | 12 | **20** | 0 | 0 | 0 | 0 | 0 |
| blocks | 18 | **35** | **35** | **35** | 21 | 33 | 27 | **35** | **35** |
| childsnack | 0 | 0 | **1** | 0 | 0 | 0 | 0 | 0 | 0 |
| data-network | 0 | 1 | **5** | 2 | 0 | 0 | 0 | 0 | 0 |
| depot | 4 | 14 | **18** | **18** | 6 | 5 | 5 | 15 | 13 |
| driverlog | 7 | **19** | 18 | 18 | 8 | 12 | 13 | 15 | 14 |
| elevators | 0 | 5 | **20** | 7 | 0 | 0 | 0 | 0 | 0 |
| floortile | 0 | 0 | **2** | 1 | 0 | 0 | 0 | 0 | 0 |
| freecell | 20 | 46 | 79 | **80** | 18 | **80** | 23 | 61 | 57 |
| ged | 0 | **20** | **20** | **20** | 0 | **20** | 0 | 0 | 0 |
| grid | 1 | 3 | 4 | **5** | 0 | 3 | 0 | 3 | 4 |
| gripper | 8 | **20** | **20** | **20** | 8 | **20** | 10 | **20** | **20** |
| hiking | 2 | 3 | **20** | **20** | 2 | 7 | 5 | 3 | 3 |
| logistics | 2 | 7 | **29** | 15 | 2 | 2 | 3 | 6 | 5 |
| maintenance | 0 | **14** | 11 | **14** | 0 | 0 | 0 | 0 | 0 |
| miconic | 55 | **150** | **150** | **150** | 71 | **150** | 146 | **150** | **150** |
| movie | **30** | **30** | **30** | **30** | **30** | **30** | **30** | **30** | **30** |
| mprime | 20 | 21 | **32** | 22 | 5 | 18 | 13 | 18 | 17 |
| mystery | 15 | 15 | **17** | 14 | 9 | 6 | 12 | 10 | 9 |
| openstacks | 0 | 0 | 2 | **20** | 0 | 1 | 8 | 2 | 5 |
| organic-synthesis | **3** | **3** | 2 | **3** | **3** | **3** | **3** | **3** | 2 |
| parcprinter | 0 | 12 | **20** | 18 | 0 | 0 | 1 | 0 | 0 |
| parking | 0 | 0 | **7** | 0 | 0 | 0 | 0 | 0 | 0 |
| pathways | 4 | 5 | **10** | 8 | 4 | 4 | 4 | 4 | 4 |
| pegsol | 17 | **20** | **20** | **20** | 18 | **20** | **20** | **20** | **20** |
| pipesworld | 12 | 22 | 23 | **27** | 12 | 13 | 15 | 20 | 17 |
| psr | 49 | **50** | **50** | **50** | **50** | **50** | **50** | **50** | **50** |
| rovers | 6 | 21 | **26** | 25 | 6 | 14 | 16 | 16 | 16 |
| satellite | 6 | 15 | **27** | 12 | 7 | 15 | 8 | 8 | 9 |
| scanalyzer | 4 | **20** | 18 | **20** | 5 | 18 | 11 | **20** | 18 |
| snake | 3 | 4 | 5 | **7** | 4 | 3 | 3 | **7** | 6 |
| sokoban | 6 | 13 | **19** | 10 | 8 | 11 | 12 | 10 | 9 |
| spider | 1 | 12 | 9 | **19** | 0 | 5 | 7 | 5 | 9 |
| storage | 14 | 18 | **19** | **19** | 17 | 15 | **19** | 18 | **19** |
| termes | 0 | 10 | 14 | **15** | 3 | 4 | 2 | 7 | 3 |
| tetris | 0 | **20** | 9 | **20** | 2 | 11 | 19 | 2 | 18 |
| thoughtful | 5 | 5 | 8 | **14** | 5 | 12 | 5 | 5 | 5 |
| tidybot | 3 | 19 | 16 | **20** | 0 | 1 | 7 | 8 | 2 |
| tpp | 6 | 13 | 23 | **29** | 6 | 15 | 10 | 14 | 15 |
| transport | 0 | 5 | 0 | **16** | 0 | 0 | 0 | 0 | 0 |
| trucks | 6 | 9 | **15** | 9 | 6 | 6 | 7 | 7 | 7 |
| visitall | 0 | **20** | 0 | **20** | 0 | 0 | 9 | 0 | 0 |
| woodworking | 1 | 1 | 2 | **4** | 1 | 1 | 1 | 1 | 1 |
| zenotravel | 8 | **20** | **20** | **20** | 7 | 9 | 9 | 13 | 14 |
| SUM | 359 | 775 | 933 | **965** | 346 | 651 | 558 | 631 | 632 |
| $>h_{gc}$ | | | | | 4 | 6 | 4 | 6 | 7 |
| $>h_{ff}$ | | | | | 2 | 5 | 5 | 4 | 3 |
| $>h_{lm}$ | | | | | 0 | 2 | 1 | 0 | 0 |

Table 3: Instance-Specific Learning: IPC Benchmark Instances, 8GB, 30min total limit (including training data generation, training $h_{nn}$, and search using $h_{nn}$) per instance: Coverage Results. In the bottom 3 rows, "$>h_{heuristic}$" indicates the count # of domains with higher coverage than $h_{heuristic}$.