# OpenReview forum: "Learning Search-Space Specific Heuristics Using Neural Network"
_icaps-conference.org/ICAPS/2020/Workshop/HSDIP — HSDIP 2020_

### Official Review · AnonReviewer3 · 2020-08-13
**Interesting approach, but carelessly written**

**Rating:** 5
**Confidence:** 4

**Review:**


== SUMMARY ==

The authors proposed a new method to learn heuristics for a single planning instance which builds upon LHCFP (Geissmann 2015) and other recent developments in learning and planning. Their network input is the state represented either in SAS or STRIPS encoding. Training states are generated by explicit backwards search or regression using Random Walks or a hill climbing approach. They train very small networks which quickly predict heuristic values. In general, they evaluate their method on the IPC tasks with a time limit of 30 minutes for training and search. Thus,  the training time has pay off already on the first search. They show that they produce results competitive to hFF and hLM.


== FEEDBACK ===

I think the authors propose interesting an promising modifications for LHCFP. Experimenting with explicit search from some goal state and regression and using hill climbing are interesting ideas. I was especially impressed how fast the small networks evaluate (and that they are quicker than hFF). At the moment they do not outperform hFF and hLM on all IPC domains, but they show already promising results on some domains.

We already assumed that the Boolean representation performs better than a plain SAS representation, because the SAS representation somehow randomly groups facts into variables (depending on the mutex groups in a task, different assignments from facts to variables are possible) and the plain SAS representation implies a wrong numeric dependency between the facts of the same variable. It is good to see that someone evaluated this.

After reading the paper, an important question is for me, what is your forward heuristic? It sounds like some standard heuristic to guide your hill climbing, but you state you do not need standard heuristics. And if it is a standard heuristic, which one have you chosen? The choice of such a heuristic could have a huge impact on how far the hill-climbing gets away from the goal.

I want to point out that the inverse actions are a hack for your explicit search. A fact in the delete list of an action does not need to be present in the state on which the action is applied. A fact present in the add list of an action does not need to be missing the the starting state. The inverse action might still be a good hack, but I kindly ask you to add this information.

Figure 4 looks very nice, but cannot show what you want to show us. Especially, you state that it shows the high accuracy of SING for state with heuristic values smaller than 30. I cannot verify this, because the data points are over-painted by the data points of the other algorithms. After a few data points have been plotted on the same (or overlapping) locations, the color intensity does not change anymore and we cannot see if there are 5 or 100 data points with a deviation of 20. Furthermore, if we have multiple different algorithms (like here), the data points of the later plotted algorithms hide the data points of the earlier algorithms. Shifting the dots plotted for each algorithm by a small value, is not a solution. It does not solve the first problem, that we do not know how many data points are at a single location.
A possible solution is to use boxplots. For every distance (or for binned distances) and every algorithm add a boxplot. Every algorithm gets boxplots in a different color (e.g. SING is blue) and the boxplots of the n-th algorithm is slightly shifted on the x axis by n * x, such that they can never hide each other.

From my background the paper is easy to comprehend, but contains too many writing errors. Typos in general, typos in references, many inline citations contain typos (e.g. 'Geissmann (2015', missing '.', missing year, missing 'et al.'). Furthermore, the paper does not use the AAAI20 style (see section titles and citation style).


Overall I think the paper makes an interesting contribution, with promising results and I would like to accept it and to discuss it during the workshop. Sadly the submitted paper is very poorly written. Especially the inline citations contain a lot of errors. It is so poorly written (see == Typos and Style and Citations (!) ==) that I cannot convince myself to accept it.





== QUESTION ==
- page 3: You state that SING expands the goal to state by randomly assigning values to the unassigned variables. The Fast Downward translator might have detected many mutex groups and used only a subset of them as SAS+ variables (because they overlap). The unused mutex groups are still available in Fast Downward. Does your goal extension also takes those into consideration?
- page 4/8: You state that you do not need a standard heuristic for training data generation. But you use a forward heuristic in the data generation. What is this heuristic?
- page 7: What do you mean by "with less training samples of 100k"?
- page 7, table 3: Some IPC benchmarks have multiple different versions (e.g. transport exists for optimal and satisficing planning and for different years). Which versions have you used for those benchmarks with multiple versions?



== MINOR ISSUES ==
- figure 1-3: In the plots you use "lmc"/"SING" instead of "hLM"/"hNN". Please make this consistent.
- figure 1-3:
	- The left column is not aligned with the right column
	- The title is redundant and too long. For the left column use just "expansions" and for the right one "search time". The domains are known from the caption of the figure, the heuristics are known from the labels of the axis.
	- Remove the "(solved .../wins...)" from the labels of the axis. This makes labels unnecessary complicated. Why can we remove "solved"? We see how many data points are on the unsolved lines of the plots. Why can we removed the "wins"? The wins just counts who is faster, but we see this already in the plot. Furthermore, the plot shows whether the wins are by a small or large fraction.
	- The caption states that we compare "run-time", the subplot title says "search time". Chose one and be consistent.


== Typos and Style and Citations (!) ==

- frequently: every 'hff', 'hlm' occurrence is missing a white space afterwards. I guess you use a command like \hff. Then you have to write \hff{} otherwise, the blank after the command is swallowed.
- frequently: missing '.' after 'et al'
- frequently: missing year after inline citation.
- page 1: "a instance-specific" -> "an instance-specific"
- page 1: "by Geissmann (2015" -> "by Geissmann (2015)"
	(I use the following command for inline citations: \newcommand{\inlinecite}[1]{\citeauthor{#1} \shortcite{#1}})
- page 2: "share a single share space" -> "share a single state space"
- page 2: "with hff and hlm". hff and hlm are not defined. Please define them the first time you use them. State their full name and citation. You use them frequently, but just define them at the end.
- page 2: "Applying a in s, s transitions to [...]" -> "After applying [...]"
- page 2: "a 5-tuple, $\PI = \langle V, O, I, G, \rangle". Either you are missing your fifth item (after the "G,") or it is a 4-tuple.
- page 2: "Ferber et investigated" -> "Ferber, Helmert, and Hoffmann (2020) ..." or "Ferber et al. (2020) ...". Use command for inline citation let it automatically use the right format.
- page 2: "using a heuristic as h ff" -> "using a heuristic as h_{ff}"
- page 2: "Arfaee et al and" -> "Arfaee et al. (YEAR)"
- page 2: "and Lelis evaluated" -> "Lelis et al. (YEAR)"
- page 3: "Although regression is useful perform backward search, [...]" -> This sentence is strange. Please reformulate.
- page 3: "Fast-Downward" -> "Fast Downward" (it is two words without a hyphen)
- page 4: "We identify var.-value pairs" -> "We identify variable-value pairs"
- page 5: "[Hoffmann, ]" -> defect citation. Missing year (same in the reference at the end)
- page 6: "from IPC" -> "from the IPC."
- page 6, table 2: The table is fine. I think it might look better, if you use instead of vertical (|) bars a little bit of space to separate the blocks (add an additional 'p{Xcm}' column), use '\cmidrule{x-y} to add horizontal lines spanning the different blocks (i.e. one line spanning hnn -> hlm for coverage, one for median # expansions, ...).
- page 7: "Ferber et al compute" -> "Ferber et al. (2020)"
- page 7: "investigated by Ferber et al (2020)" -> "Ferber et al. (2020)"
- page 7, Table 3: Use either "hlm" or "lmc". Make this consistent.
- page 7: "Shen et al (2020)" -> "Shen et al. (2020)"
- page 8: "[Fern, Yoon, and Givan, 2004]]" -> "[Fern, Yoon, and Givan, 2004]"
- page 9: "functions inclassical planning" -> "functions in classical planning"
- page 9: "[Hoffmann, ]" -> ?
- page 9: "Ff domain collection" -> "FF domain collection"

---

> ### Author Response · Authors · 2020-08-28
> **Response to the questions**
>
> Thank you so much for your reviewing. Here is our answers.
> (Also, as submitted in the common response, please be noticed we have updated a new version of our paper with the writing mistakes revised.  https://rb.gy/5xiors)
>
> -----
> [Question] page 3: You state that SING expands the goal to state by randomly assigning values to the unassigned variables. The Fast Downward translator might have detected many mutex groups and used only a subset of them as SAS+ variables (because they overlap). The unused mutex groups are still available in Fast Downward. Does your goal extension also takes those into consideration?
>
> [Answer] Although we implemented several methods of exploiting such unused mutex groups, they did not perform well in preliminary experiments so the experiments in the paper uses the simple random assignment method. We are continuing to investigate mutex group based assignments.
>
> -----
> [Question] page 4/8: You state that you do not need a standard heuristic for training data generation. But you use a forward heuristic in the data generation. What is this heuristic?
>
> [Answer] Although the search away from a goal during the training data generation phase can use a heuristic evaluation function as indicated in Sec 4.2,  we chose to use  blind search (i.e., no heuristic) in the experiments reported in the paper, as we are initially interested in seeing how far a system which learns from scratch without the aid of any existing heuristics can be pushed.   In future work, we’ll investigate the use of heuristics during the search during training data generation including bootstrapping (using a previously learned h_nn) or a standard heuristic function such as h_ff.
> We’ll change the description in 4.2 (the description in Step 4) to clarify that in the experiments in this paper, no heuristic function is used.
>
> -----
> [Question] page 7: What do you mean by "with less training samples of 100k"?
>
> [Answer] As shown in Table 1, “samples #” column, the original configuration of C5 samples 400k states, which is 4 times as many as the other configurations in Table1. In the ablation study we use C51, which is the same as C5 except that the number of samples is 100K, which the same as the number of samples for C2, C3, C4, to eliminate the sample # difference between the C5 variants and C2/C3/C4 and facilitate comparison. We’ll add this explanation to the revision.
>
> ----
> [Question] page 7, table 3: Some IPC benchmarks have multiple different versions (e.g. transport exists for optimal and satisficing planning and for different years). Which versions have you used for those benchmarks with multiple versions?
>
> [Answer] All of the IPC benchmarks are from the satisficing track. The specific versions used are listed below. We’ll clarify the domain versions in the revision.
> agricola-sat18-strips
> barman-sat14-strips
> childsnack-sat14-strips
> data-network-sat18-strips
> elevators-sat11-strips
> floortile-sat14-strips
> ged-sat14-strips
> hiking-sat14-strips
> logistics98
> maintenance-sat14-adl
> openstacks-sat14-strips
> organic-synthesis-sat18-strips
> parcprinter-sat11-strips
> parking-sat14-strips
> pegsol-sat11-strips
> pipesworld-tankage
> scanalyzer-sat11-strips
> snake-sat18-strips
> sokoban-sat11-strips
> spider-sat18-strips
> termes-sat18-strips
> tetris-sat14-strips
> thoughtful-sat14-strips
> tidybot-sat11-strips
> transport-sat14-strips
> trucks-strips
> visitall-sat14-strips
> woodworking-sat11-strips

---

> > ### Comment · AnonReviewer3 · 2020-09-04
> > **Response & Updated Score**
> >
> > Thank you for your clarifications. The new uploaded version looks way better, but still contains incorrect citation and does NOT use the AAAI20 style!
> >
> > I quickly looked through the pages, thus, this is not an exhaustive list:
> >   - page 2: '[Ferber ...] investigates' -> 'Ferber ...(YEAR)' investigates
> >   - page 2: 'Geissman investigated' -> 'Geissman (YEAR) investigated'
> >
> > I still ask you to update your figure 4, because almost all data points of SING are hidden for heuristic values < 30. Thus, we cannot reason about them.
> >
> > It seems we cannot update our scores. I increase my score by one.
> > 6: Marginally above acceptance threshold

---

### Official Review · AnonReviewer1 · 2020-08-16
**Interesting paper, but needs some work on the presentation**

**Rating:** 6
**Confidence:** 4

**Review:**

The paper describes and evaluates on a standard benchmark set neural-network
heuristic functions learned from samples obtain by backward search.

It is an interesting paper showing that even a very simple type of neural
networks can learn useful heuristic function if it is fed with a suitable data.
In this case the data was obtained by two variants of backward search and it
seems to me that the generation of (full) states in a backward search was the
main difficulty the authors had to deal with.

In my opinion, the paper probably warrants publication on the workshop, but I
I have some reservations about the paper:
1. Regarding the presentation: The authors use two variants of a problem
representation, STRIPS and FDR, which, sometimes, makes the presentation
confusing. For example in Section 4.2, the paragraph "Regression" starts with
saying that "modified SAS+ representation" used by Alcazar et al. (2013) is
used here and then follows a desription using a STRIPS action. First, this way
you introduce another formalism, because Alcazar et al. is using SAS+
formalism with preconditions, effects, and *prevail conditions* which is the
main difference to FDR. Second, if you choose to describe the conditions with
STRIPS actions, why do you need to talk about another formalism?

My suggestion would be to use only one formalism (in this case FDR would be
the better choice, because you use variable encoding for the neural network),
and reformulate the whole paper in this formalism (which is certainly
possible). It could help if you define a set of facts in FDR as a set of all
variable-value pairs, e.g., F = {(V,v) | V \in Vars, v \in Dom(V)}, and allow
interpretation of preconditions and effects as sets of facts.

2. Regarding the method: If I understood correctly, in the "regression"
variant, you treat the undefined variables as if none of the facts from the
corresponding variable is set, which seems to me incorrect (although it seems
that it works well in the experiments), because undefined variable means that
there can be any value of the variable (or one can use mutexes to infer which
of the facts can be in a reachable state). Can you explain why did you choose
to interpret it this way? It is not clear to me from the text.


Minor issues:
First page, last paragraph: missing parenthesis after "by Geissmann (2015",
and extra comma after h_{nn}.

Section 3, second paragraph "Ferber et" -> "Ferber et al."

Section 5, the item starting with "blocks: " -- invalid citation

It seems there is a missing space after every subscript all over the text
(like after h_nn, h_ff and so on).

---

> ### Author Response · Authors · 2020-08-28
> **Response to the questions**
>
> Thank you so much for your reviewing. Here is our answers.
> (Also, as submitted in the common response, please be noticed we have updated a new version of our paper with the writing mistakes revised. https://rb.gy/5xiors)
>
> ------
> [Question] 1. Regarding the presentation: The authors use two variants of a problem representation, STRIPS and FDR, which, sometimes, makes the presentation confusing. For example in Section 4.2, the paragraph "Regression" starts with saying that "modified SAS+ representation" used by Alcazar et al. (2013) is used here and then follows a desription using a STRIPS action. First, this way you introduce another formalism, because Alcazar et al. is using SAS+ formalism with preconditions, effects, and prevail conditions which is the main difference to FDR. Second, if you choose to describe the conditions with STRIPS actions, why do you need to talk about another formalism?
> My suggestion would be to use only one formalism (in this case FDR would be the better choice, because you use variable encoding for the neural network), and reformulate the whole paper in this formalism (which is certainly possible). It could help if you define a set of facts in FDR as a set of all variable-value pairs, e.g., F = {(V,v) | V \in Vars, v \in Dom(V)}, and allow interpretation of preconditions and effects as sets of facts.
>
> [Answer] First, we’d like to clarify a possible misunderstanding -- although previous work (LHCFP) used only the SAS+ encodings in the neural network, we investigated both SAS+ and STRIPS encodings. As indicated in Table 1, configuration C3 uses the SAS+ encoding, but configurations C1, C2, C4, and C5 use the boolean (STRIPS) representation. Overall, the configurations using the STRIPS representation performed better, and our ablation study in Section 6.1 shows that the boolean representation has a significant effect on performance.
> Next, regarding the multiple formalisms: We thought that STRIPS needed to be introduced because as explained above,  it is the representation used in the most successful SING configurations, and also because  our inverse action generation strategy for the explicit backward search in training data generation (Sec. 4.2) exploits the STRIPS representation.
> The modified SAS+ formalism by Alcazar et al. was mentioned because the backward search code is implemented as an extension of Fast Downward using Alcazar et al’s representation. However, we agree that the presentation could be clearer, and in the revision, we’ll try to carefully review and clarify our usage of formalisms.
>
> ------
> [Question] 2. Regarding the method: If I understood correctly, in the "regression" variant, you treat the undefined variables as if none of the facts from the corresponding variable is set, which seems to me incorrect (although it seems that it works well in the experiments), because undefined variable means that there can be any value of the variable (or one can use mutexes to infer which of the facts can be in a reachable state). Can you explain why did you choose to interpret it this way? It is not clear to me from the text.
>
> [Answer] First, the last sentence of paragraph 2 in Section 4.2 ("When generating training data, undefined values are treated as if all STRIPS propositions represented by the mutex group (SAS+ group) are false.") was incorrectly written.
> Although the regression planner based on Fast Downward uses undefined values, the neural network which we are training is for forward search, and the forward space state does not have undefined values.
> When generating training data, a bit vector representation of the SAS+ states needs to be generated. Thus, when converting the SAS+-based representation used by Fast Downward into a bit vector for the training data, unlike the other possible state values in the mutex group, undefined values are not explicitly represented in the bit vector. For example, suppose a state variable v in regression search has 2 possible actual values, val1 and val2, as well as "undefined".  In the bit vector representation output for use as training data,  v  is represented by 2-bits, where the first bit represents val1, and the second bit represents val2, and there is no explicit third bit for the undefined value.
> We will correct and clarify Section 4.2 as above.

---

### Author Response · Authors · 2020-08-28
**Updated revised version**

First, we apologize for the numerous writing errors and citation-related bugs. Much of the text was re-written immediately before the submission deadline and we were unable to proofread adequately.  All of the errors pointed out by both reviewers have been corrected in the current version. You can download the latest version through this link: https://rb.gy/5xiors

---

### Author Response · Authors · 2020-09-05
**A new version with more improvements**

Please find another updated version of the paper at https://rb.gy/yyfilu
We have changed Figure 4 to a boxplot as suggested by Reviewer1.
As suggested by Reviewer2, we have cleaned up the use of multiple formalisms.
STRIPS is no longer mentioned in the preliminaries section, and is introduced in Section 3.1 (many of the sections have been renumbered since the original submission) when describing the state representation used by the neural net.
In addition, many improvements have been made in the text for conciseness and improved clarity.

---

### Comment · Program_Chairs · 2020-09-14
**Final Decision: Accept**

Dear Authors,

Thank you very much for your submission. We are happy to inform you that we have decided to accept it and we look forward to your talk in the workshop. You will receive additional information per mail in the coming days.

Best,
The HSDIP'20 team

---

### Decision · Program_Chairs · 2020-09-30

Accept